# Methods for dealing with unequal cluster sizes in cluster randomized trials: A scoping review

Denghuang Zhan [1,2‡]*, Liang Xu[1,2‡], Yongdong Ouyang[1,2‡], Richard Sawatzky[2,3], Hubert Wong[1,2]

**1** School of Population and Public Health, University of British Columbia, Vancouver, British Columbia, Canada, **2** Centre for Health Evaluation and Outcomes Sciences, University of British Columbia, Vancouver, British Columbia, Canada, **3** School of Nursing, Trinity Western University, Langley City, British Columbia, Canada

‡ These authors contributed equally to this work and share first authorship.
* jzhan@cheos.ubc.ca, zdhjeffrey@gmail.com

## Abstract

In a cluster-randomized trial (CRT), the number of participants enrolled often varies across clusters. This variation should be considered during both trial design and data analysis to ensure statistical performance goals are achieved. Most methodological literature on the CRT design has assumed equal cluster sizes. This scoping review focuses on methodology for unequal cluster size CRTs. EMBASE, Medline, Google Scholar, MathSciNet and Web of Science databases were searched to identify English-language articles reporting on methodology for unequal cluster size CRTs published until March 2021. We extracted data on the focus of the paper (power calculation, Type I error etc.), the type of CRT, the type and the range of parameter values investigated (number of clusters, mean cluster size, cluster size coefficient of variation, intra-cluster correlation coefficient, etc.), and the main conclusions. Seventy-nine of 5032 identified papers met the inclusion criteria. Papers primarily focused on the parallel-arm CRT (p-CRT, n = 60, 76%) and the stepped-wedge CRT (n = 14, 18%). Roughly 75% of the papers addressed trial design issues (sample size/power calculation) while 25% focused on analysis considerations (Type I error, bias, etc.). The ranges of parameter values explored varied substantially across different studies. Methods for accounting for unequal cluster sizes in the p-CRT have been investigated extensively for Gaussian and binary outcomes. Synthesizing the findings of these works is difficult as the magnitude of impact of the unequal cluster sizes varies substantially across the combinations and ranges of input parameters. Limited investigations have been done for other combinations of a CRT design by outcome type, particularly methodology involving binary outcomes—the most commonly used type of primary outcome in trials. The paucity of methodological papers outside of the p-CRT with Gaussian or binary outcomes highlights the need for further methodological development to fill the gaps.

**Data Availability Statement:** All relevant data are within the paper and its S1 Appendix, S1, S2 Files.

**Funding:** The authors disclosed receipt of the following financial support for the research, authorship, and/or publication of this article: This

work was supported by the BC SUPPORT Unit [grant number RWCT-204]. The funders had no involvement in study design, collection, analysis and interpretation of data, reporting or the decision to publish.

**Competing interests:** The author(s) declare no potential conflicts of interest with respect to the research, authorship, and/or publication of this article.

## 1. Introduction

A cluster-randomized trial (CRT), also known as a group-randomized trial, is an experimental design for comparing treatment effects by randomly assigning clusters (groups) of participants to different treatments, wherein all members of a given cluster receive the same treatment. In specific contexts, these trials have also been called community-intervention trials or place-based trials. Common types of CRTs include: parallel-arm CRT (p-CRT), stepped-wedge CRT (sw-CRT), partially nested CRT (pn-CRT), and cross-over CRT (co-CRT). CRTs are often used when individual randomization is not practical for logistical or cost reasons, or would lead to potential contamination of treatment groups due to interaction between participants or providers of the intervention [1–3].

The key feature of a CRT is that the responses from individuals within a cluster are likely more similar than with those from different clusters. This is because individuals within a cluster may share similar characteristics or be exposed to the same external factors associated with membership in a particular cluster [3, 4]. It is well-known that analytic approaches that do not account for within-cluster correlation underestimate standard errors (SEs) of intervention effects and inflate Type I errors [3]. The most widely used approaches for analyzing clustered data are: (generalized) linear mixed effects models ((G)LMM), marginal/generalized estimating equation (GEE) models, and cluster-level analyses (i.e., modeling cluster-level summaries obtained by aggregating individual-level data).

A pair of recent articles by Turner *et al.* [5, 6] reviewed recent methodological developments for the design and the analysis of CRTs, respectively. Murray *et al.* [7], Campbell *et al.* [8], Rutterford *et al.* [9] and Gao *et al.* [10] summarized a wide range of sample size calculation methods available for p-CRT while Baio *et al.* [11] provided a methodology review on sample size calculations specifically for the sw-CRT. Most literature investigating CRT design methods have assumed all clusters have the same number of participants.

In practice the assumption of equal cluster sizes seldom is met. In many trials, the cluster will comprise groups of patients being served by organizational units such as communities, hospitals, or clinics which naturally vary in size. Even when a trial plans to enroll equal numbers in each cluster, unequal cluster sizes may arise as a result of variability in the availability of participants and differential dropout rates across clusters. Hence, in real trials, the number of clusters may be low and clusters typically have unequal, and sometimes very low, numbers of participants [12–14]. Exact formulae for obtaining desired statistical properties (such as bias, Type I error rates, power, etc.) are available only under very restricted conditions (e.g., multivariate normal outcomes with equal cluster sizes), and typically, asymptotic approximations must be used. However, when cluster sizes are unequal and potentially small, calculations based on these asymptotic approximations may be inaccurate and this issue should be considered during both the trial design as well as the data analysis stages.

Typically, Type I error inflation occurs when statistical methods appropriate for balanced data are applied to data with unequal cluster sizes [15, 16]. This problem may be exacerbated when the number of participants in some clusters or the number of clusters is small. When cluster sizes are variable, the use of the mean cluster size in the simple design effect will underestimate the required sample size, more so as the variation in cluster sizes increases. Use of the maximum cluster size as an alternative may be overly conservative. Eldridge *et al.* [17] concluded that when the coefficient of variation (CV) of the cluster sizes, defined as the ratio of the standard deviation of the cluster sizes to the mean cluster size, is above 0.23 then unequal cluster sizes should not be ignored in the p-CRT. Strategies to correct sample size calculations for unequal cluster sizes include evaluating relative efficiencies or design effects based on the cluster size CV.

It should also be noted that the trial power calculated using standard software is an expected value, which averages the results over all allocations the randomization algorithm can generate. In a CRT with unequal cluster sizes, the attained power (i.e., the power calculated conditional on the allocation generated by the randomization) varies across allocations. Emerging work by Martin *et al.* [18], Wong *et al.* [19] and Matthews [20] have shown the importance of considering not just the expected trial power, but also the risk of getting an unacceptably low attained power.

This paper is a scoping review of the literature focusing on methods for dealing with unequal cluster sizes in the design and analysis of CRTs. Previous literature reviews have identified various key papers (e.g., 11 papers in Turner *et al.* [5], 8 papers in Turner *et al.* [6]), but no systematic scoping review appears to have been published. This scoping review is intended to identify: (1) the impact of unequal cluster sizes on the statistical properties in both the design (i.e. power, sample size, etc.) and the analysis (i.e. Type I error rate, bias, coverage, etc.) phases, (2) the existing methods for dealing with unequal cluster sizes, and (3) the gaps in current knowledge and informing directions for development of new methods in this area. Our intent is to assist readers in identifying literature relevant to the design and analysis of their CRTs when cluster sizes are unequal.

## 2. Materials and methods

We conducted this review in three stages, based on the scoping review guidelines by Arksey *et al.* [21], by: (1) identifying the relevant papers by electronic databases, reference lists, and hand searching of key journals; (2) selecting the relevant papers based on the inclusion criteria and extracting the information of interest from the papers; (3) summarizing and reporting the results.

### 2.1 Searching

We searched Medline, EMBASE, Google scholar, Web of Science and MathSciNet databases from inception to Feb 23$^{th}$, 2021. The searches were limited to documents written in English. The search strategy for MEDLINE and EMBASE is detailed in the S1 Appendix.

Briefly, we started with Medline and EMBASE as the primary databases and then used Google Scholar, Web of Science Core Collection and MathSciNet as auxiliary search engines. The vocabulary used to describe properties of analytic models (which emphasizes issues such as type of model, bias, Type I error, and coverage) is distinct from that used to describe properties of the design (which emphasizes power or sample size, often working with an analytic model that is implicit and not always fully described). Hence, for the search in Medline and EMBASE, we structured the search to cover the design and analysis considerations separately. We first extracted the keywords and terms from several methodology papers involving unequal cluster sizes initially identified through preliminary Google Scholar searches [17–19, 22–25]. We then identified additional terms with similar meanings and grouped them to capture papers focused on either design methodology or analysis methodology, based on a keywords search (Fig 1). To be specific, for design methodology (Branch A), we searched for papers with a keyword list that included both (1) a well-known type of CRT (the left circle), and (2) a term related to unequal cluster sizes or relevant to trial design (the right circle). For analysis methodology (Branch B), we searched for papers with a keyword list that included (1) an analysis method commonly used for CRTs and (2) a term referencing cluster or sample size, and (3) a term related to statistical performance. The final result is the union of Branches A and B. To identify other potentially relevant papers, we also examined the reference lists of all relevant papers, systematic reviews identified by our searches, and the authors' personal files through

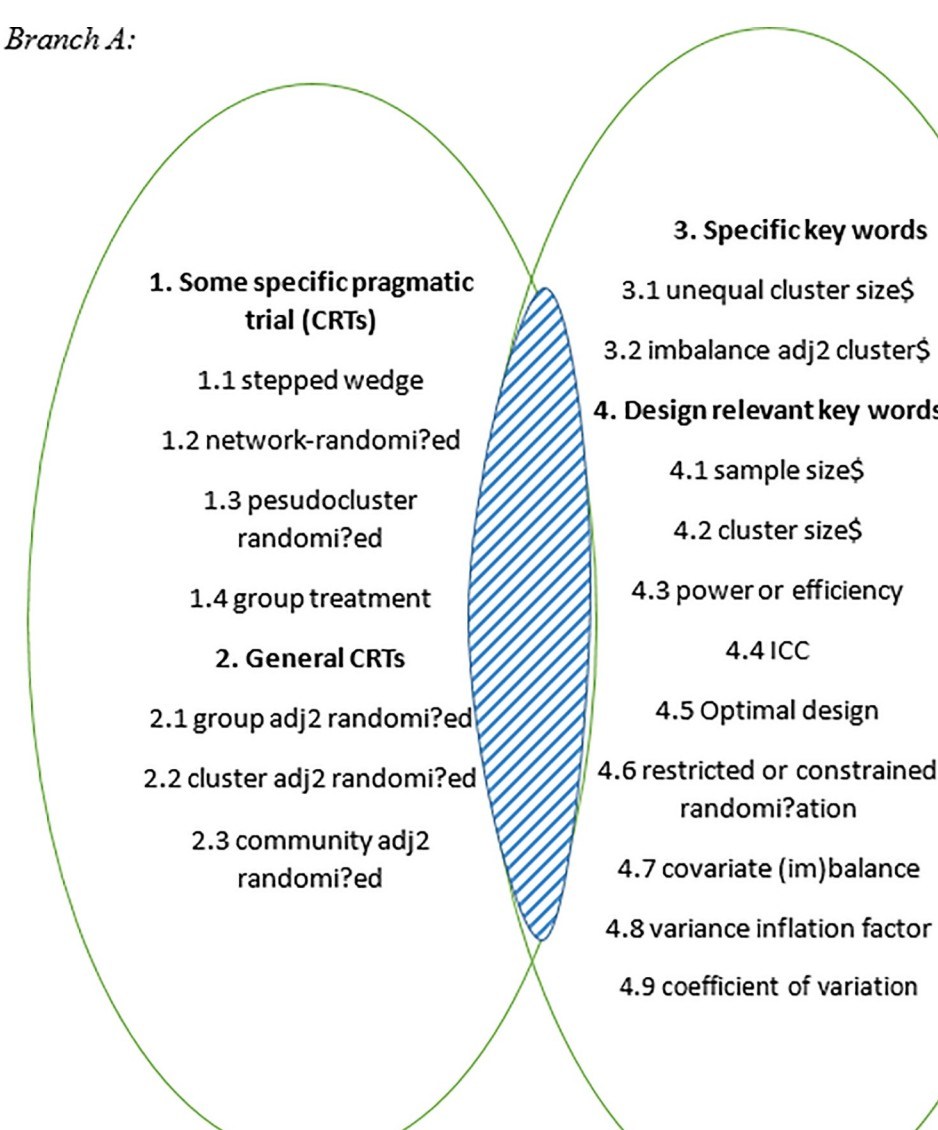

**Branch A:**

**Branch B:**

**Fig 1. Two branches indicating the search strategies in Medline and EMBASE.**

MathSciNet or Google Scholar. We concluded with backward and forward searches based on several key articles identified in the previous steps.

Branch A: Focus on design methodology. The left oval captures the union of terms (numbered 1 and 2) used to describe cluster randomized trials. The right oval captures the union of terms used to refer to unequal cluster sizes (number 3) or are relevant to trial design (number 4). The search result is the intersection of the two ovals. The operator "adj2" is used to capture instances where the two words are separated by a third word (e.g., cluster is randomized). Branch B: Focus on analytic methodology. The large rectangle represents terms used to describe analysis methods commonly used for CRTs (number 5. The left small oval (term 6.1) represents keywords referencing cluster or sample size and right small oval (term 6.2) contains keywords related to statistical performance. The search result is the intersection of the rectangle and two ovals.

## 2.2 Study selection and data extraction

A paper was considered as relevant if it involved methodology for unequal cluster sizes in any type of CRT and reported on either: (1) design considerations (e.g. impact on or calculation of sample size/power) or (2) analytic performance (e.g. Type I error, bias, etc.). Any publication type was considered as long as the paper met the inclusion criteria. We however excluded studies that referenced pre-existing methodology for unequal cluster sizes but did not investigate (e.g. through analytical formula derivation or simulation) the statistical performance of such methods.

Three reviewers (DZ, LX, YO) independently reviewed the title and abstract of each paper identified by the Medline and EMBASE search engine based on the inclusion criteria above. Any uncertainties were discussed with clarifications made to the inclusion criteria as needed, as per scoping review methods. Full-text articles were retrieved if the inclusion criteria were met or if the abstract did not contain sufficient information; full texts were screened in duplicate by two independent reviewers (DZ and LX) and the disagreements were resolved through discussion including other authors (HW, YO).

Data were extracted by one reviewer (DZ) and checked by a second reviewer (LX) and a third reviewer (YO). Extraction was guided by a template matrix developed for this review and approved by all authors, which included data on: citation details (authors, paper title, publication year, etc.), focus of the paper (power calculation, Type I error etc.), trial characteristics (type of CRT design, cluster sizes, CV, etc.), the statistical models used, and the main findings.

## 3. Search results

From the main search of EMBASE and Medline, 73 papers met the search criteria. Six additional papers were identified through backward and forward citation searches conducted on six key papers through Google Scholar, Web of Science and MathSciNet. The search terms and strategies used in the two main search databases (Medline and EMBASE) captured all relevant papers from the citation searches, except for six papers that were published in journals that are not fully indexed by those two search engines [26–31]. In total, we identified 79 relevant papers. Fig 2 summarizes the results of the search process.

The earliest publication was from 1981 and 50 (63%) appeared in 2010 or later. The types of trial addressed were: 60 (76%) on the p-CRT, 14 (18%) on the sw-CRT, four (5%) on the pn-CRT and one (1%) on the co-CRT. The type of outcomes considered were: 52 (54%) continuous, 33 (34%) binary, 8 (8%) count and 3 (3%) survival. The analytic models considered were: 41 (45%) mixed effects, 26 (28%) GEE, and 25 (27%) other. The ranges of values for the trial parameters were: 6 to 500 for the number of clusters, 0 to 2.1 for the cluster size CV, and 0 to

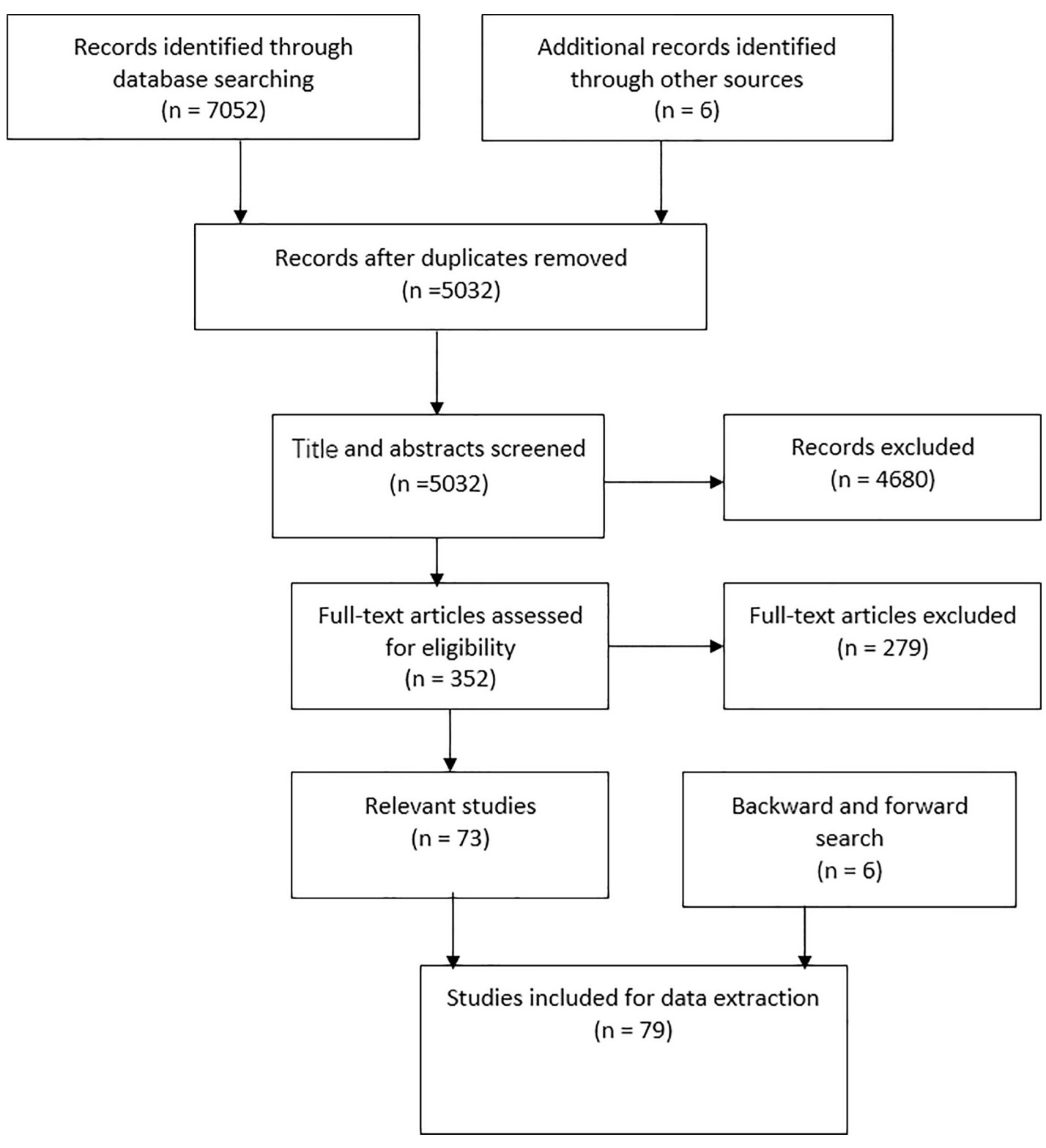

**Fig 2. The flow chart of the search process.**

**Table 1. Paper information matrix for different types of CRT designs and outcomes.**

| Trial design | Outcome measure | Model | Analysis (Type I error, bias and coverage) | Design | | |
|---|---|---|---|---|---|---|
| | | | | Sample size/power via simulation | Relative efficiency (RE) corrections | Design effect (DE) and related corrections |
| **Parallel arm CRTs** | Continuous | GEE | [12, 43, 53] | [12, 35, 63] | [22, 43, 44] | [17, 27, 28, 62] |
| | | Mixed effects | [12, 30, 32, 37, 51–55] | [12, 18, 19, 23, 32, 35, 48, 50, 63] | [29, 33, 39, 41, 96] | [37, 48, 50, 51, 62] |
| | | Cluster level | | [12] | | [17] |
| | | Other type | | [34, 38] | [33, 40, 42] | [26, 36, 46, 47] |
| | Binary | GEE | [43, 68–71, 97] | [60, 68] | [22, 43, 44] | [17, 89] |
| | | Mixed effects | [54, 66, 98] | [23, 60, 66, 98] | [58] | [49, 62] |
| | | Cluster level | | | | [17] |
| | | Other type | [31, 67] | [56, 57, 99, 100] | [82] | [36, 47, 59, 61, 64] |
| | Count | GEE | [43, 72, 74] | [72] | [22, 43, 44] | |
| | | Mixed effects | [72] | [72] | | [49] |
| | | Other type | | | | [73] |
| | Survival | Other type | [76, 77] | [75–77] | | [75, 76] |
| **Stepped wedge CRTs** | Continuous | GEE | [24] | [24] | | [25] |
| | | Mixed effects | [24, 80] | [18, 19, 24, 83] | [82, 84, 85] | [20, 25] |
| | Binary | GEE | [87–89] | [88] | | |
| | | Mixed effects | | | | [86] |
| | Count | Mixed effects | | | [84] | |
| **Partially nested CRTs** | Continuous | Mixed effects | | [90] | [92, 93] | [91] |
| **Crossover CRTs** | Binary | GEE and other types | | | | [94] |

0.99 for the intra-cluster correlation coefficient (ICC), defined as the ratio of between-cluster variance to the total variance.

Table 1 summarizes the characteristics of the papers according to the type of the trial design, the type of outcome, the analytic model used, and the focus (design or analysis). To facilitate easy identification of the papers that would be relevant in a specific application, we have provided separate S2 File for each subsection below describing the trial characteristics (CV, ICC, etc.) that were investigated in each paper.

## 3.1 Parallel-arm CRT

In the p-CRT, each cluster is randomly allocated to one intervention at the start of the trial and every participant in the cluster enrolled throughout the trial receives that intervention. Among the 60 papers discussing the p-CRT, 19 (32%) focused on analysis considerations (Type I error rate, bias, etc.) while most of the papers (n = 52, 87%) discussed how unequal cluster sizes impact trial design and proposed methods for the sample size/power calculations using inflation factors applied to results for equal cluster size designs, through simulation, or derivation of approximate analytic formulae. In the following sections, we first classified the paper based on the type of the outcome, i.e., continuous (Gaussian), binary, count or survival. For continuous and binary outcomes, we further classified the papers based on whether they focused on design or analysis as the majority of the papers (~91%) focused on these two types of outcomes.

**3.1.1 Continuous outcomes.** In total, we identified 33 papers addressing continuous outcomes; 31 papers investigated design considerations, and eight investigated analysis

considerations. (See S1 Table in S2 File) Linear mixed effects models were most commonly used for the analysis, though about a quarter of the papers considered GEEs.

*3.1.1.1 Design considerations.* Multiple studies found that increasing cluster size variability led to decreasing power [32–35], and that the magnitude of the power loss depended on the value of the ICC [33, 36]. Eldridge *et al.* [17] concluded that if the cluster size CV is less than 0.23, there is no need to account for the effects of variable cluster size in the sample size calculation. Guittet *el al.* [37] showed that relatively extreme cluster size imbalance (e.g. when cluster sizes followed a Pareto distribution) was associated with higher sensitivity to misspecification of the ICC and a corresponding loss in power. In order to compensate for the power loss, various papers have proposed adjustments to the sample size formulae for equal-cluster size trials based on design effects or relative efficiency considerations. In addition, Carter [38] suggested constraining the randomization to reduce sample size imbalance across the treatment arms and so mitigate the magnitude of the power loss.

*3.1.1.1.1 Relative efficiency (RE) of unequal versus equal cluster sizes.* Various authors have studied the relative efficiency (RE) of unequal versus equal cluster size designs and proposed adjustments to compensate for the loss of power. Candel and Breukelen have done extensive work on the RE and their work mainly focused on moderate cluster sizes (mean from 20 to 55) with low to moderate variation in the cluster size (CV from 0.07 to 0.80). Specifically, Candel *et al.* [29] studied the *D*-optimality of equal vs. unequal cluster sizes, and used simulations to evaluate to what extent the asymptotic results on RE hold, with an ICC ranging from 0.01 to 0.3 and an average cluster size of 24. Based on the maximum loss of efficiency with $D_s$- and/or *D*-criterion, the authors recommended inflating the number of clusters by 18% and 22% for estimating variance components and fixed effects, respectively. Subsequently, they found that for small sample sizes, the asymptotic REs were very accurate for the $D_s$-criterion of the fixed effects and even more accurate for the *D*-criterion. Breukelen *et al.* [39] presented a maximum likelihood-based Taylor approximation to the RE as a function of the mean and variance of cluster size and the ICC. A minimum RE value could thus be obtained which depended on CV only. They concluded that for most cases, increasing the number of clusters by 11% would compensate for the loss of efficiency. They further suggested adjusting for the efficiency loss due to cluster size variation by multiplying the number of clusters by $1/RE_{min}$ where $RE_{min} = (1-CV^2/4)$. Breukelen and Candel [40] also compared one RE formula proposed by Manatunga *et al.* [26] and two RE formulae proposed by Breukelen *et al.* [39] together and claimed that the efficiency loss due to varying cluster size in cluster randomized trials was smaller than had been suggested previously in the literature. Breukelen and Candel [41] further examined the expected efficiency loss due to varying cluster sizes based on the asymptotic RE, as well as the accuracy of the asymptotic RE and its Taylor approximation for small sample sizes. However, they found in empirical testing that these formulae could overestimate the empirical RE, so provided a table for the maximum overestimation of the RE in selected scenarios to facilitate adjustment.

You *et al.* [42] suggested approximating the RE using the non-centrality parameter in the t-test and proposed using this non-centrality parameter to adjust the mean cluster size or both the mean cluster size and the number of clusters for continuous outcomes; however, this approach was not validated empirically. Liu and Colditz [22] derived an analytic RE formula for continuous, binary, and count outcomes in GEE models, computed the RE under various patterns of cluster size imbalance, and proposed a sample size adjustment procedure to account for the efficiency loss. Subsequently, Liu *et al.* [43] derived the RE of two bias-corrected sandwich estimators based on GEE for continuous, binary, or count outcomes for when the number of clusters is small and they proposed an algorithm for increasing the number of clusters to compensate the efficiency loss due to the unequal cluster sizes when the CV is

known and unknown. Liu and Colditz [44] then extended these two bias-corrected sandwich estimators to three-level models and again proposed increases in the number of clusters to compensate for the efficiency loss. Innocenti *et al*. [45] compared three two-stage sampling designs which allow cluster size to vary in the population, and derived the optimal sample sizes for each design under a budget constraint. By comparing the relative efficiency of the three optimal two-stage sampling designs, they recommended the sampling clusters with probability proportional to size design as the most efficient one in their setting.

**3.1.1.1.2 Design effects and adjusted corrections.** The design effect (DE), sometimes also called the variance inflation factor (VIF), refers to the ratio of the variance of an effect estimate from a cluster randomized trial to the variance of that effect estimate from an individually randomized trial with the same number of participants. Donner *et al*. [46] showed that multiplying the sample size calculated for the individually randomized trial by the DE yields the required number of individuals in the CRT, assuming equal cluster sizes. Several DEs have been derived under various conditions and assumptions to account also for cluster size variability.

Manatunga *et al*. [26] showed that when the cluster size CV is modest (0.35–0.47), the required total sample size can be obtained by adding a correction term to the traditional formula based on the average cluster size. Wang *et al*. [27] extended Manatunga *et al*.'s [26] work and derived a closed-form sample size formula for size-stratified CRTs with random cluster sizes (discrete uniform and zero-truncated negative binomial distributed, CV = 0.509, 0.351, 0.279) at the design stage. Kerry and Bland [47] compared design effects calculated using three different weightings (equal, cluster size, or minimum variance) of the cluster means and found the minimum variance weights minimized the required sample size. Similarly, Guittet *et al*. [37] derived an adaptation of the VIF from the minimum variance weights correction to be used in case of the cluster size imbalance. The simulations confirmed that the minimum variance weights correction of the VIF used in the sample size calculations maintained nominal power even when the CV was relatively large (0.1). Hemming *et al*. [36] also proposed a VIF correction that can be applied in the situation when the number of clusters is fixed in advance for both binary and continuous outcomes. Finally, Mukaka and Moulton [28] compared empirical power from three different definitions of the DE (based on the arithmetic mean, the harmonic mean, and the squared CV ($CV^2$) of cluster sizes) and different analytic models in a scenario with a CV of around 0.53. They concluded that the performance of the different sample size methods depended on the assumed analytic model.

**3.1.1.1.3 Other approaches.** Various other methods for addressing key considerations in the design of CRTs have been proposed. Shi and Lee [23] developed a method for sample size calculation for unequal cluster sizes based on Monte Carlo simulation in the mixed effects model framework. This approach incorporates the variation of cluster sizes and can be applied to CRTs with both binary and continuous outcomes. Two papers considered CRT designs with longitudinal outcomes. Heo *et al*. [48] proposed sample size calculation methods based on ordinary least squares (OLS) estimates for detecting an intervention by time interaction with random slopes. Their simulation results indicated that the analytical power based on OLS was comparable to the empirical power based on (restricted) maximum likelihood estimates even with varying cluster sizes (uniform distributed with cluster sizes mean ranging from 5 to 25). Amatya and Bhaumik [49] also derived sample size formulae based on the (G)LMM framework for continuous, binary and count outcomes. The proposed formulae maintained nominal Type I error rates and target power well with unequal cluster sizes (distributed uniformly between 10 and 30, or from a Gamma distribution with shape 4, scale 5) and differential attrition rates over treatment groups.

Two papers considered sample size re-estimation procedures. Lake *et al*. [50] proposed a two-step sample size re-estimation approach for correcting an initial sample size calculation

that assumed a constant cluster size. Nuisance parameters (including cluster size variability) are estimated from an internal pilot study and the sample size is then adjusted with another sample size formula that accounts for cluster size variation. In a simulation study (8 or more clusters, truncated negative binomial distributed cluster sizes with mean 4 with a CV of 0.5), the sample size re-estimation method was shown to rescue studies that would have been underpowered given the initial sample size calculation, with minimal Type I Error inflation. Harden and Friede [51] proposed two sample size recalculation procedures based on re-estimating nuisance parameter values from interim data using mixed effects models, allowing for unequal cluster sizes. Their simulation results showed that the proposed unadjusted estimator maintained the type I error rate and the power close to the nominal levels while the bias-adjusted estimator exhibited some type I error rate inflation.

*3.1.1.2 Analysis considerations.* A common finding among the identified papers was that Type I error rates increase as cluster sizes variability increases. However, the magnitude of impact varies across combinations of the analytic model, the number of clusters, the ICC, and the average cluster size, so choosing an analysis that achieves nominal Type I error rates is not straightforward. The most comprehensive investigations were conducted by Johnson *et al.* [52] and Leyrat *et al.* [12]. Johnson *et al.* [52] investigated Wald tests from two-level linear mixed effect models (i.e. "one-stage" analysis) with either the "Between-Within" or the Kenward-Roger (KR) denominator degrees of freedom, with or without constraining variance components to be positive, and cluster-level models (i.e. "two-stage" analysis) with various weighting schemes. In simulations with varying numbers of clusters (2 to 16) per arm, mean cluster sizes (8 to 128), and ICCs (0.001 to 0.1), they concluded that for studies with at least 6 clusters per arm, the cluster-level model weighted by the inverse of the estimated theoretical variance of the cluster means, with the estimates constrained to be positive, had the best Type I error control, and that for studies with at least 14 clusters per arm, the one-stage model with the KR degrees of freedom and unconstrained variance estimation performed well. However, the optimal choice varied across settings and they recommended that users working with a specific setting could consult their online tables. The range of CV values (which were not reported by the authors) appeared to be relatively narrow from 0 to approximately 0.40. Leyrat *et al.* [12] extended Johnson et al.'s work by considering also the Satherwaite denominator degrees of freedom in the mixed effects model, the Wilcoxon rank sum test in the cluster-level analysis, and GEEs with or without small-sample corrected robust standard error (SE) estimators. The ranges of input parameter values were: ICCs—0.001 to 0.05, numbers of clusters—4 to 200), and cluster size CV—0.4 to 0.8. How well each method maintained nominal Type I error rates varied across the combinations of parameter values. Generally, they found that (in agreement with Johnson et al.) the variance-weighted cluster-level analysis performed well across the scenarios considered. The mixed model with the Satherwaite degrees of freedom performed well when there were at least 10 clusters, and the GEE with small-sample corrections performed well when there were at least 30 clusters. Feng *et al.* [53] investigated the performance of GEE (without small sample corrections), maximum likelihood based on linear mixed model, bootstrap, and a four-stage method on estimating the SEs of individual covariate effects and the treatment by covariate interaction. Under limited simulations with 10 clusters per arm, a cluster size CV of 0.78, and ICCs of 0.1 and 0.5, they found that both the bootstrap method and (in agreement with the works discussed above) the GEE underestimated these SEs. While maximum likelihood and the four-stage method performed comparably, the latter was recommended as it tended to be more conservative. Additional results have been reported for these and other analytic approaches in more limited settings. Guittet *et al.* [37] showed through simulations that an imbalance in cluster sizes did not result in biased restricted maximum likelihood (REML) estimates with the linear mixed effects model. However, extreme

imbalance (e.g. when cluster sizes followed a Pareto distribution) was associated with inflated Type I error and power loss.

Agbla *et al.* [54] demonstrated that the use of two-stage least squares regression applied to cluster-level summaries was valid for intention to treat (ITT) analysis for treatment effect in CRTs where non-adherence occurs at either the cluster level or the individual level, for both continuous and binary outcomes. They found that if the trial has highly imbalanced cluster sizes (Pareto distributed with mean 20, shape 1.8, scale 9.1), the application of a small-sample degrees of freedom correction can maintain the nominal coverage. When cluster sizes were very imbalanced, the Huber-White correction yielded appropriate SEs, but the point estimates could still be biased. The authors also advised against the use of cluster size weights when cluster sizes are highly imbalanced. Finally, Hermans *et al.* [30, 55] used sample-splitting and pseudo-likelihood to derive closed-form estimators for mixed effects model with unequal cluster sizes and showed that their proposed methods were as efficient as maximum likelihood estimation but needed less computation time.

**3.1.2 Binary outcomes.** In total, we found 30 papers addressing binary outcomes; 26 papers investigated design considerations, and 10 investigated analysis considerations. (See S2 Table in S2 File) Unlike the work on continuous outcomes where use of mixed effects models was predominant, work on binary outcomes was split roughly equally between mixed effects models and GEEs.

*3.1.2.1 Design considerations*. The effect of the unequal cluster sizes on power for binary outcomes have been widely investigated under various parameters settings. As was seen with continuous outcomes, several papers found that the power was not greatly impacted when the cluster size variability was modest (e.g., $CV < 0.3$) [58, 59], but could be appreciable when the CV was large [60]. Chakraborty *et al.* [56] proposed a simulation-based approach to estimating ICC when cluster sizes vary for subsequent sample size calculation. Chakraborty *et al.* [57] later showed via simulation that the ICC value depended upon the event rate and event rate variations in addition to the cluster size, cluster size variation, and the number of clusters. Multiple approaches to adjusting the sample size formulae for unequal-cluster size trials based on design effects or relative efficiency considerations have been proposed, that each of unique applications.

*3.1.2.1.1 Relative efficiency (RE) of unequal versus equal cluster sizes.* Candel and Breukelen [58] first derived the asymptotic RE based on first-order approximation to the marginal quasi-likelihood (MQL), and then determined via simulation, for CVs in the range 0.31 to 0.71, the correction factors for converting the variance of this MQL estimator into the variance of the second-order penalized quasi-likelihood (PQL) estimator to obtain the sample size adjustment formula through second-order PQL estimation. RE-based corrections based primarily on the sandwich estimator in GEEs have also been developed; see 3.1.1.1 for discussion of the work by Liu and colleagues [22, 43, 44].

*3.1.2.1.2 Design effects and adjusted corrections.* Similar to Kerry and Bland's [47] work, Jung *et al.* [59] proposed a sample size calculation method using the minimum variance weights. Braun [60] proposed a method for sample size calculation based on mixed models using a saddlepoint approximation to the distribution of the cluster means while accounting for the correlation between ICC and cluster size, and showed their approach was similar to the first-order asymptotic approximations when variation in clusters sizes was small but was more conservative with larger variation in cluster sizes. Kong [61] proposed an adjusted DE which was more accurate than the maximum cluster size and the average cluster size methods to prevent the power loss when there is a relatively large number of clusters (30~500) and ICC is high (0.25–0.75). Similar to Kong, Eldridge *et al.* [17] proposed a corrected design effect based on maximum possible inflation for sample size calculation with a CV ranging from 0.65 to

0.95. Taljaard *et al.* [62] proposed sample size formulae to account for potential attrition in CRTs by incorporating the values of 'missing data mechanism ICC', which quantified the variability in follow-up rates. Caille *et al.* [63] derived an analytical formula for power based on a GEE approach for dichotomized continuous outcomes, assuming equal cluster sizes, and conducted simulations with unequal cluster sizes to evaluate its accuracy. Dichotomization was hypothesized to increase power by reducing ICC, but it led to an overall loss of power, because information was lost through dichotomization. For a stratified cluster randomization design, Xu *et al.* [64] presented closed-form sample size formulae for estimating the required total number of subjects and for estimating the number of clusters (12 to 147) for each group per stratum, based on the Cochran-Mantel-Haenszel statistic. Kennedy-Shaffer and Hughes [65] proposed analytic formulae for the sample size calculation for stratified designs and the ratio of the sample sizes for stratified vs unstratified CRTs with binary outcomes.

*3.1.2.2 Analysis considerations.* Overall, the methods/models of choice depend on the feature of the trial such as the CV, number of clusters etc. The most comprehensive studies were conducted by Li and Redden on the generalized linear mixed model (GLMM) [74] and the GEE [73], covering ranges for the number of clusters (10 to 30, GLMM; 10 to 20 GEE), average cluster size (20 to 100, GLMM; 10 to 150 GEE), ICCs (0.001 to 0.1, GLMM; 0.001 to 0.05, GEE), and cluster size CVs (0 to 1). For the GLMM, they concluded that the "Between-Within" approximation to the *F*-test denominator degrees of freedom maintained nominal test size well, while the KR and Satterthwaite approximations tended to be overly conservative. For the GEE, they recommended that to maintain nominal test size, the analysis should be based on a *t*-test with the Kauermann and Carroll correction to the robust SE when there is moderate variation in the cluster sizes (CV<0.6) and with the Fay and Graubard correction when there is large variation in cluster sizes (CV>0.6), but cautioned that no single estimator was universally superior.

The investigations of other papers were much more limited in scope or addressed specialized questions. Heo and Leon [66] evaluated via simulation the performance of maximum likelihood estimation (MLE) in the mixed effects logistic regression model with equal and unequal cluster sizes in terms of Type I error rate, power, bias, and SE. Across varying treatment effects, number of clusters (20 to 400), mean cluster sizes (5 to 100, uniformly distributed) and ICC values (0 to 0.5), the results showed that the performance of the mixed effects logistic regression model with cluster size CVs ranging from 0.029 to 0.35) was similar to its performance with equally sized clusters. Stefanescu and Turnbull [67] investigated maximum likelihood estimation with saturated and unsaturated models when cluster sizes were not equal and proposed using the EM algorithm to obtain maximum likelihood estimates. Stedman *et al.* [68] used simulation to compare several methods for estimating power, coverage, bias, and mean squared error. They found that when the cluster size varies (1 to 150) and the number of clusters is relatively large (458), the bootstrapped methods had higher mean squared error and were less powerful compared with GEE and full MLE methods. Penalized quasi-likelihood (PQL) method yielded biased results and the permutation test preserved Type I error rates, but had less power than the other methods considered. Westgate and Braun [69] found that with cluster size imbalance and cluster-level covariates, the quadratic inference functions (QIF) and a modified version of QIF that has estimating equations in the same class as GEE were both less efficient than GEE assuming an exchangeable correlation structure, for small (20) and large (100) numbers of clusters. Mancl and DeRouen [70] proposed an alternative covariance estimator to the robust covariance estimator of GEE. They compared the bias-corrected covariance estimator to the robust, model-based and degrees-of-freedom-adjusted and jack-knife covariance estimators for binary responses. They pointed out that the bias-corrected covariance estimator using an *F*-distribution rather than the asymptotic Chi-square distribution gave tests with sizes close to the nominal level even with as few as 10 clusters with unequal

sizes (mean cluster size = 32), whereas the robust and jackknife covariance estimators gave tests with sizes that could be 2–3 times the nominal level. The degrees-of-freedom-adjusted estimator with the F-distribution performed well only when cluster sizes are equal, while the performance of the model-based estimator depended on the correct specification of the correlation structure.

Hoffman *et al.* [31] suggested using within-cluster resampling (WCR) for analyzing data with varying cluster sizes that are informative in the context of the binary outcome, but noted that the theory is also applicable to other types of outcome. Also addressing informative cluster sizes, Williamson *et al.* [71] proposed a less computationally intensive approach that uses GEE weighted inversely with cluster size. They showed that their approach was asymptotically equivalent to WCR but less biased when sample sizes are small.

**3.1.3 Count outcomes.** We identified seven papers addressing count outcomes (See S3 Table in S2 File); see 3.1.1.1 for discussion of the papers by Liu and colleagues [22, 43, 44]. Of note, Pacheco *et al.* [72] assessed the performance of five analysis methods (the cluster level two-sample t-test, GEE with empirical covariance estimators, GEE with model-based covariance estimators, GLMM and Bayesian hierarchical model) for count data in p-CRTs in terms of coverage, Type I error rate, power and random-effects estimation when there were variations in cluster size (CV = 0.2 and 0.6 with a mean of 30). They concluded the imbalance impacted the overall performance (a decrease in power, higher coverage than nominal level and higher widths) of the cluster-level t-test as well as the GEE's coverage in small samples (with 10 or 20 clusters). Wang *et al.* [73] extended their formula for continuous outcomes [27] to count outcomes and showed that the proposed method maintained the nominal power and Type I error rate well. Li *et al.* [74] proposed a sample size formula which incorporated pragmatic features such as arbitrary randomization ratio, over-dispersion, random variability in cluster size, and unequal lengths of follow-up with count outcomes based on the GEE. Their simulation results showed that the proposed sample size method was robust to unequal cluster sizes (Uniform distributed from U(34,56) or U(10,80)) and that larger variability in cluster size leads to larger sample sizes.

**3.1.4 Survival outcomes.** We identified only three papers focusing on survival outcomes (See S4 Table in S2 File). Unlike CRTs with continuous, binary or count outcome, regular models such as (G)LMM and GEE will not be suitable for CRTs with survival outcome. Jung [75] first proposed a simulation-based sample size calculation method using the weighted rank test for a survival outcome which allows for variable cluster sizes. The choice of optimal weight function is complex as it depends on clustered survival time's correlation structure, the alternative hypothesis on two marginal survival distributions, and the cluster size distributions. Li and Jung [76] proposed a closed form sample size formula for weighted rank tests which depend on the mean and the variance of cluster size distribution. Stedman *et al.* [77] used simulations to compare the nonlinear mixed model, the marginal proportional hazards model with robust SEs, the clustered log-rank test, and the clustered permutation test for analyzing clustered survival data from physician-randomized trials of health care quality improvement interventions with up to 458 clusters when there is a variation in cluster size. They found that Type I error was underestimated for the clustered log-rank test but was preserved for the other methods. Nonlinear mixed models performed best in terms of power, bias and coverage but only when the distribution of the random effect was correctly specified.

## 3.2 Stepped wedge CRTs

In the sw-CRT design, all clusters begin by delivering the standard of care treatment. Each cluster is randomized to transition to delivering the new treatment at one of several pre-

specified later time-points so that by the end of the trial, all clusters are delivering the new treatment. Reasons for using the sw-CRT design include: (1) the objective is to assess the effectiveness of implementing the new treatment in the population, (2) the number of clusters which can switch at one time-point is limited (e.g. due to lack of resources), and (3) clusters may refuse to participate without certainty that they ultimately will be switched to the new treatment [24, 78]. The sw-CRT and the p-CRT share similar methodological concerns, but there is additional complexity for the sw-CRT due to the presence of confounding of treatment effects with time effects. This requires adjustment for time effects in the analytic model which strongly impacts power. However, since the correlation structure is more complex, sample size adjustment cannot be easily achieved by applying simple design effects [79]. Investigation of the unequal cluster size sw-CRT design is an area of current research with 13 of the 14 papers identified having been published in the past five years.

**3.2.1 Continuous outcome.** We identified 10 papers addressing continuous outcomes; all papers investigated design consideration, and one investigated analysis considerations (See S5 Table in S2 File). Hussey and Hughes [24] considered unequal cluster sizes with the CV ranging from 0 to 1 in a trial with a continuous or binary outcome. They observed greater power when the data were analyzed using the GEE or the GLMM instead of a cluster-level analysis based on a LMM. They suggested this was due to the cluster-level analysis not weighting observations optimally in the presence of unequal cluster sizes. Hughes *et al*. [80] further derived a closed-form expression for an estimate of the intervention effect together with its SE based on a permutation argument. They showed that these estimates are robust to misspecification of the mean structure, the covariance structure and the distribution of the data-generating mechanism. The simulation studies showed that even when the cluster size varies between clusters (with average cluster size of 10) according to a lognormal distribution with standard deviation of 0.2 (low variation) or standard deviation of 1.0 (high variation), the proposed estimate was unbiased, and maintained the nominal Type I error rate and coverage well, while power declined as cluster size variance increased. The authors also warned that as the estimate is derived based on identity link, it will be a biased estimate of the intervention effect from an individual-level model if the link function is nonlinear. Based on the DE developed by Woertman *et al*. [81] to calculate sample size for the equal-cluster sized sw-CRT from the sample size for the p-CRT, and work by Eldridge *et al*. [17] and Kerry and Bland [47] for the unequal cluster size p-CRT, Kristunas *et al*. [25] proposed two adjusted DEs for obtaining the sample sizes in the unequal cluster size in sw-CRT from the calculation for an individually randomized trial. The two adjusted DEs are derived based on the cluster weights and minimum variance weights, respectively. They compared empirical power obtained using the DE derived by Woertman *et al*. [81] and these two proposed adjusted DEs, with varying degrees of imbalance (CV from 0 to 1.689). The simulated data were analyzed by GEE. They found that the imbalance in cluster size did not notably affect power of sw-CRTs and that use of the two proposed adjusted DEs resulted in an over-powered design. Girling [82] compared the RE of single period p-CRTs with sw-CRTs and showed how the methods used to assess experimental precision for single period p-CRTs with unequal cluster sizes can be extended to stepped wedge and other complete layouts under longitudinal or cross-sectional sampling for binary outcomes with a large range of CV (from 0.315 to 1).

Martin *et al*. [18] and Wong *et al*. [19] showed that the 'attained' power corresponding to a given allocation in a stepped-wedge trial could be substantially lower than the expected power when the variability in cluster sizes was large (CV > 0.7). The authors of both papers noted that constraining the randomization (e.g. transition the clusters with large sizes at the first or last steps to reduce the correlation between participant treatment allocation and time) would reduce the risk of a trial achieving inadequate power. Ouyang *et al*. [83] further explored the

relationship between 'attained' power and two allocation characteristics: the correlation between treatment and time at individual-level, and the absolute treatment group imbalance. Through simulations, they found that attained power was strongly affected by the treatment-vs-time correlation but very little by the absolute treatment group imbalance. Matthews [20] developed a regression-based approximation formula to the variance of the treatment effect which can be used to identify the allocations that yield high precision for cross-sectional designs, and extensions were made for closed-cohort designs. Furthermore, Harrison *et al*. [84] investigated the RE of the sw-CRT with unequal cluster sizes to equal cluster sizes and derived three analytical formulae for sample size/power estimation when different types of information (e.g., cluster sizes, order of randomization) are available. They also proposed a method to identify the upper and lower bounds of the 'attained' power. Their formulae also extended the Hussey and Hughes mixed effects model to allow for random time effects. The formulae can be applied to both continuous and count outcomes. Their simulation results showed that increasing the ICC (from 0.01 to 0.25) and the CV (from 0 to 2) both decrease the RE. Kasza *et al*. [85] investigated the information content of sequence-period cells when cluster sizes are unequal. Based on the mixed effects model, they demonstrated that the efficiency of incomplete designs depends on which cells are excluded, rather than purely on sample size.

**3.2.2. Binary outcome.** We identified six papers addressing binary outcomes; 4 papers investigated design consideration, and investigated analysis considerations (See S6 Table in S2 File). For binary outcome, Zhou *et al*. [86] investigated numerically the impact of variable cluster size (CVs of 0.6 and 2) on the power with binary outcomes based on mixed effect model. They found that both the CV and the study participant intervention-control allocation ratio affected the study power. In terms of applying GEE to binary outcomes, Li *et al*. [87] proposed an efficient estimating equations approach to analyze cluster-period means which requires less computation time and showed that when the number of clusters is limited and cluster sizes are unequal, the proposed matrix-adjusted estimating equations can still substantially improve the coverage of the correlations parameters. Both Thompson *et al*. [88] and Ford and Westgate [89] conducted large simulation studies to compare small-sample standard-error corrections such as Fay and Graubard (FG); Mancl and DeRouen (MD); Kauermann and Carroll (KC) etc. for GEE when cluster sizes vary and the number of clusters is small. Specifically, Thompson *et al*. [88] suggested that when the number of clusters is smaller than 50, the KC-approximation method, or FG standard errors with an independent working correlation structure are recommended, while KC-approximated standard errors with an exchangeable working correlation structure are recommended if cluster sizes are large (300). FG standard errors with an exchangeable working correlation structure are recommended if cluster sizes are large and ICC is low (0.01). Ford and Westgate [89] showed similar results with Thompson *et al*. [88], and demonstrated that in order to maintain nominal type I error rates, certain combinations of bias corrections to standard error estimates and degrees of freedom approximations are recommended depending on the specific scenario.

## 3.3 Partially nested randomized control trials

The pn-CRT is a trial in which outcomes are independent in one arm but are correlated in the other arm, and thus is a hybrid of an individually randomized control trial and a cluster randomized trial. A pn-CRT design is common in educational and behavioral research where the new intervention often is delivered in a group setting (outcomes are clustered), whereas the standard of care in the control arm involves no additional clustering (independent outcomes). In these trials, individuals in the intervention arm typically are randomized individually, not by group, but the statistical calculations to account for the clustered outcomes are the same.

The general conclusions and findings for pn-CRT are similar to p-CRT. We identified four papers in total for pn-CRT and all of them focused on continuous outcomes and used mixed effect models (See S7 Table in S2 File).

Roberts and Roberts [90] showed that failing to consider cluster size variation in the design and analysis of a pn-RCT could underestimate the required sample size. Hedges and Citkowicz [91] showed how information on the ICC can be used to correct for bias in the treatment effect and its variance when the cluster sizes are equal. They suggested that in the case of the unequal cluster sizes, the treatment effect estimator is unchanged while for the calculation of the variance estimator, the mean cluster size can be used. Candel and Breukelen [92] derived asymptotic RE of unequal vs equal cluster sizes for pn-CRT with a small cluster size mean (6 or 10) and a medium CV (0.24 to 0.55) and concluded that efficiency loss due to unequal cluster sizes is usually less than 10 percent. They suggested inflating the sample size in each arm by 11 percent to compensate. They also derived the asymptotic RE of unequal versus equal cluster size designs in terms of the $D$-criterion and $D_s$-criteria for maximum likelihood estimation in a mixed effects linear regression with similar ranges for the cluster size mean and the CV [93].

## 3.4 Crossover CRTs

In the co-CRT for comparing two interventions, the trial is divided into two time periods. Each cluster is randomized to deliver one intervention for the first period and crosses over (switches) to delivering the other intervention for the second period. The co-CRT is particularly efficient in settings where only a limited number of clusters are available for study, as it provides within-cluster comparisons of the intervention effect.

Our search identified only one paper addressing unequal cluster sizes in the co-CRT. For binary data, Forbes *et al.* [94] derived formulae for sample size estimation for different estimators and unbalanced cluster sizes by first assuming the number of individuals per period is constant within each cluster, but not across clusters and then describe the extension when the numbers of individuals vary across both periods and clusters. Four different estimators (unweighted, cluster-size-weighted, inverse-variance weighted at the cluster-level and GEE with an independent correlation structure at the individual-level) were evaluated in a simulation study. With small to large cluster sizes (mean from 50 to 1000, 6–50 clusters), small ICC (0.01, 0.05 and 0.1) and small to large CV (0, 0.40, 0.70, 1), the power calculated using their sample size formulae was comparable to the simulated power. No systematic bias was reported for any of the estimators, but coverage was low when the baseline risk was low (less than 6%) and the within-cluster-within-period correlation was at least 0.05. The three cluster-level estimators performed comparably well although the inverse-variance weighted estimator was optimal. The GEE estimator however had noticeably lower efficiency and power when the cluster sizes were large (1000) and unbalanced across periods.

## 4. Discussion

Unequal cluster sizes in cluster randomized trials are common and the papers identified in this review showed that failing to consider the potential effects of unequal cluster sizes during both trial design and data analysis could introduce estimation bias, increase Type I error rates, or decrease power/efficiency.

We found that most papers addressing unequal cluster sizes focused on the parallel-arm CRT (~80%) and were published within the last 20 years (~80%). Multiple studies have investigated the impact (primarily with continuous or binary outcomes) across various ranges of ICC, CV, number of clusters, and average cluster sizes on the performance of the most commonly used models (i.e. mixed effects, GEE, cluster level). The impact of unequal cluster sizes

on trial performance characteristics varied widely across values of the input parameters, and ultimately, whether one should account for unequal cluster sizes depends on what is considered an acceptable error if the results from assuming equal cluster sizes were used. We did not find any papers synthesizing the findings across the diverse scenarios that have been investigated. Note that the variation in the true Type I error rates of different statistical tests complicates the comparison of the power values from methods based on different tests, because differences in power in part may be due to differences in the true Type I error rates. Further development is needed for outcomes that are not continuous or binary.

For the unequal cluster size sw-CRT, the limited number of papers identified indicates many methodological gaps need to be filled. The available papers focused almost exclusively on design considerations for trials with continuous or binary outcomes and only a few works addressed the performance (the coverage, Type I error rate etc.) of the analytic models. Of note, sample size calculation formula for binary outcomes have not been published. Our search identified only one paper discussing count outcomes and no papers dealing with time to event (survival) outcomes. The impact of design features (number of steps, number of clusters, number of clusters switching at each time point, etc.) as well as population characteristics (time effects, ICC, etc.) on the power needs to be examined under variation in cluster sizes, when the outcome is non-Gaussian. In particular, the impact of the high correlation between the treatment status and time in the sw-CRT on power warrants further investigation. Outside of the p-CRT and the sw-CRT, the literature on the handling of unequal cluster sizes in other types of CRTs is scant.

Although review of statistical software was not part of this work, we noted that computational results presented in the papers reported on here were obtained using SAS (80%), R (16%), Stata (3%) and Python (1%). Regarding data analysis, the models (mixed effects, GEE, cluster-level analysis) used can be fit using standard software packages. However, software for sample size/power calculations is lacking. For some of the papers, portions of the code were provided in S2 File or were indicated as being available on request. However, we found only one currently available package, 'clusterPower' in R, which allowed for unequal cluster sizes in power calculations (in the p-CRT) [26], and a Shiny CRT Calculator allowing for unequal cluster sizes in power/sample size calculation for both p-CRT and sw-CRT [95]. While additional packages are under development for sw-CRT [83, 87], the availability of user-friendly software is still lagging the methodological development and findings.

The literature on analysis of data with repeated outcome measures is vast and has been described using very diverse terminology, particularly in fields (such as psychology and ecology) outside of our focus on statistics and health. Although we included the most commonly used terms in describing cluster randomized trial methodology in our search, a limitation of this review is that relevant papers from other subject areas on the analysis of clustered data may have been missed. In addition, given the diversity of studies and their findings, an overarching synthesis with general recommendations could not be achieved. Instead, we hope that this scoping review provides readers with the information needed to enable them to quickly identify the literature most likely to be relevant to their specific trial context and to stimulate further research, with an eye towards meeting the need for guidelines and recommendations regarding unequal cluster sizes in cluster randomized trials.

## Supporting information

**S1 Appendix. The search steps in Medline and EMBASE.**
(DOCX)

**S1 File. PRISMA checklist.**
(DOC)

**S2 File. Paper information details.**
(XLSX)

## Author Contributions

**Conceptualization:** Hubert Wong.

**Methodology:** Hubert Wong.

**Supervision:** Hubert Wong.

**Writing – original draft:** Denghuang Zhan, Liang Xu, Yongdong Ouyang.

**Writing – review & editing:** Denghuang Zhan, Liang Xu, Yongdong Ouyang, Richard Sawatzky, Hubert Wong.

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
