## [Decision Letter · Decision Letter 0]

3 Feb 2021

PONE-D-20-23827

Methods for dealing with unequal cluster sizes in cluster randomized trials: A scoping review

PLOS ONE

Dear Dr. Zhan,

Thank you for submitting your manuscript to PLOS ONE. After careful consideration, we feel that it has merit but does not fully meet PLOS ONE’s publication criteria as it currently stands. Therefore, we invite you to submit a revised version of the manuscript that addresses the points raised during the review process.

We look forward to receiving your revised manuscript.

Kind regards,

Rafael Sarkis-Onofre

Academic Editor

PLOS ONE

Journal Requirements:

Reviewers' comments:

Reviewer's Responses to Questions

**Comments to the Author**

1. Is the manuscript technically sound, and do the data support the conclusions?

Reviewer #1: Yes

Reviewer #2: Partly

Reviewer #3: Yes

2. Has the statistical analysis been performed appropriately and rigorously? 

Reviewer #1: Yes

Reviewer #2: N/A

Reviewer #3: Yes

3. Have the authors made all data underlying the findings in their manuscript fully available?

Reviewer #1: Yes

Reviewer #2: Yes

Reviewer #3: Yes

4. Is the manuscript presented in an intelligible fashion and written in standard English?

Reviewer #1: Yes

Reviewer #2: Yes

Reviewer #3: Yes

5. Review Comments to the Author

Reviewer #1: The manuscript focuses on recent research studies on cluster randomized trials (CRTs) with unequal cluster sizes. The authors searched and studied the related papers from several literature databases and the inclusion steps are also presented in detail. Among the 67 papers included, the authors discussed the impacts of unequal cluster sizes on both design and analysis of cluster randomized trials by trial type. The manuscript provides a valuable resource on dealing with unequal cluster sizes for researchers working on different types of cluster randomized trials.

Major comments:

1. The authors gave a broad review on CRTs by trial type and analyzed the impact of unequal cluster sizes on both type I error and power calculation on a case-by-case basis. Each specific case, however, does no form natural transitions in terms of methods or CRT parameters. For example, the subsection of continuous outcomes in section 3.1.1 on page 10 has summarized several papers related to analysis considerations on continuous outcomes in parallel-arm CRTs with unequal cluster sizes. However, the settings, implications or conclusions vary by paper and there are no transition sentences connecting the different papers. Therefore, in addition to the similar case in the same subsection, I think grouping the papers within each subsection to describe might give more clear information and the grouping could be based on methods utilized or similar scopes. In addition, could the authors add a summarizing sentence at the beginning of each subsection to guide the readers on the specific impact of unequal cluster sizes to that subsection before describing different papers one-by-one?

2. Could the authors give slightly more information when mentioning certain number of methods? I found two: 1) on page 17, “Breukelen et al. [55] also compared three RE equations and claimed that the efficiency loss due to varying …”; 2) on page 21, “They compared empirical power obtained using the DE derived by Woertman et al. [75] and the two proposed adjusted DEs…”. Could the authors indicate the general techniques which the three and two methods are based on?

Minor comments:

1. Table 1 summarizes the methods, trial types and scopes of each paper. Could the authors also give some summary statistics of the parameters considered in the manuscript for different types of CRTs? On page 8, we have the overall ranges for some parameters such as number of clusters and it might bring more comprehensive information about parameters utilized by type for readers.

Reviewer #2: The manuscript concerns the scoping review of the literature on statistical methods in cluster randomized trial with unequal cluster sizes. By searching through EMBASE, Medline, Google Scholar, Web of Science, and MathSciNet, the authors identified relevant papers and summarized the papers in terms of trial design, outcome measure, model, and statistical ob- jectives on analysis, simulation, relative efficiency corrections, and design effect. They found that methods for the p-CRT with a Gaussian outcome with unequal cluster sizes in have been investigated quite broadly, but limited investigations have been done outside of the p-CRT with Gaussian outcome.

The main body on the results of the scoping review is in Section 3, where the authors concerns different designs in different sub sections. It seems to me this section is more like a listing of each paper described by one sentence, on what they did, rather than a “review” with some summarization and comparison.

The methods on CRTs with unequal cluster sizes were categorized to analysis (Type 1 error, bias, coverage) and design (power and size calculation), which is further described by simulations, relative efficiency corrections and design effect and related corrections, as shown in Table 1. I’m quite confused about those categories, and there seems not descriptions on how those categories are defined. In my understanding, analysis is to obtain estimate of parameter of interest related to data collected from CRTs, and design is on obtaining cluster sizes to attain desired power. Therefore, (1) I believe type 1 error is indeed a measure to evaluate study design, such that it shall be in the category of “design”; (2) simulations are always conducted to evaluate performance, in both analysis and design approaches, so I don’t understand why it shows up in “design” category parallel to RE and DE corrections.

The main searching criterion is shown in Fig 1, about which I have a few questions. The right oval in Branch A for design methodology includes key words related to unequal cluster sizes or those relevant to trial design, e.g., sample size. Therefore, based on this criterion, I would imagine a large number of papers that do not ever consider unequal cluster sizes would be included, since a very general key word (e.g., sample size) is used. I wonder why the searching criterion is set up as so. In addition, in the introduction, the authors mentioned other names for CRTs, for example place-based trials. Would this current searching criterion miss the papers with “place-based trials” indicating CRTs with unequal cluster sizes?

For Branch B for analysis methodology, the found paper would always include “random effects” or “mixed effects” or “hierarchical” or “‘generalized estimating equations” or “multi- level”, as indicated by Fig 1. I understand those are common methods to address clustered trials, however, this automatically restricts the analysis methods, and any other analysis meth- ods outside this range cannot be found.

Here are some other comments:

1. The review of papers are categorized by trial designs, i.e., p-CRT, sw-CRT, pn-CRT, and co-CRT. Even though those designs were first mentioned in the introduction, and the statistical methods for different trial designs with equal cluster sizes were discussed then, the definitions were given in Section 3. It would potentially cause trouble for readers not familiar to different trial designs. Please consider defining those designs in the introduction.

2. Page 3, line 68: “Except within a very restricted set of conditions (e.g., with multivariate normal outcomes and equal cluster sizes), exact formulae for obtaining desired statistical properties (e.g., bias, Type I error rates, power, etc.) are not available and most formulae have been derived under the assumption that the number of clusters and the sample sizes within clusters are sufficiently large for results from asymptotic theory to hold”. This paragraph is on existing reviews on statistical methods of CRTs with equal sizes. Not sure how this sentence on statistical properties depending on sufficiently large cluster numbers and sizes is relevant.

3. One of the search engines, MathSciNet, stated in Section 2.1 is not mentioned in the Abstract.

Reviewer #3: Line 20: This could explicitly state that it is the number of participants enrolled that might vary across clusters, to make it clear it is not referring to natural cluster size

Line 22: very minor point but saying CRT design could be interpreted as literature on the design of crt’s, which is not the case here as the authors are looking at design and analysis literature. See also line 66

line 37, while the authors have identified a gap in the methodological literature for alternative outcomes it would be helpful to say something about whether, from the type of trials actually being conducted, there is a need to fill this gap

line 46: after the first mention of group (cluster) could the terminology of cluster be consistently used as this is the most commonly used terminology.

Line 58. The authors introduce common methods of analysis here, it could be useful to similarly introduce sample size strategies: de, re, that are later mentioned in table 1

Line 63: the order of references 5,6 need to be switched in order for this sentence to be accurate.

Line 62: It would be helpful to describe the context or drawbacks of the 2 Turner reviews which necessitate the need for this scoping review. Were these similarly confined to p-crt? This paragraph might sit better near the end of this section to lead into the justification and scope of this review

Line 73: Could the authors provide evidence of this statement as well as the frequency of trials with low numbers of clusters.

Line 87: would be helpful to define the coefficient of variation in cluster size here for those unfamiliar

Line 106: is studies the appropriate word to describe these papers

Fig 1: was any consideration given to randomisation spelt with an s rather than z.

Line 211 is this the CV of cluster size or CV as alternative to ICC

Section 3.1.1 it would be helpful to include more of a summary here, particularly what the gaps were before delving into the details for the different outcomes.

Line 218 +: The organisation of these sections need to be improved. As a reader I would likely know the CV and ICC, number of clusters in my study and I would want to be able to quickly identify the methods that have been tested in similar conditions. This information gets somewhat lost in the text making it harder to read. Perhaps subheadings for the CV and then the ICC and cluster sizes presented at the beginning of each paragraph would help. Alternatively a summary for each analysis method (or the most common/easily implemented) and the conditions under which it works best rather than describing each paper in turn. The authors touch upon the lack of synthesis of findings in their discussion but its not clear why no synthesis has been done here as I think that is where value can be added.

Section 3.1.2 Given there are several reviews that have already considered these methods for the parallel group study is this section necessary or could it be reduced to those methods that had not been identified in previous reviews?

6. PLOS authors have the option to publish the peer review history of their article (what does this mean?). If published, this will include your full peer review and any attached files.

Reviewer #1: No

Reviewer #2: No

Reviewer #3: No

---

## [Author Response · Author response to Decision Letter 0]

22 Mar 2021

Please see the response to reviewers for details.

---

## [Decision Letter · Decision Letter 1]

11 May 2021

PONE-D-20-23827R1

Methods for dealing with unequal cluster sizes in cluster randomized trials: A scoping review

PLOS ONE

Dear Dr. Zhan,

Thank you for submitting your manuscript to PLOS ONE. After careful consideration, we feel that it has merit but does not fully meet PLOS ONE’s publication criteria as it currently stands. Therefore, we invite you to submit a revised version of the manuscript that addresses the points raised during the review process.

We look forward to receiving your revised manuscript.

Kind regards,

Rafael Sarkis-Onofre

Academic Editor

PLOS ONE

Journal Requirements:

Additional Editor Comments (if provided):

Please, review the manuscript based on reviewer 2 comments.

Reviewers' comments:

Reviewer's Responses to Questions

**Comments to the Author**

1. If the authors have adequately addressed your comments raised in a previous round of review and you feel that this manuscript is now acceptable for publication, you may indicate that here to bypass the “Comments to the Author” section, enter your conflict of interest statement in the “Confidential to Editor” section, and submit your "Accept" recommendation.

Reviewer #1: (No Response)

Reviewer #2: All comments have been addressed

Reviewer #3: All comments have been addressed

2. Is the manuscript technically sound, and do the data support the conclusions?

Reviewer #1: Yes

Reviewer #2: Yes

Reviewer #3: (No Response)

3. Has the statistical analysis been performed appropriately and rigorously? 

Reviewer #1: Yes

Reviewer #2: Yes

Reviewer #3: (No Response)

4. Have the authors made all data underlying the findings in their manuscript fully available?

Reviewer #1: Yes

Reviewer #2: Yes

Reviewer #3: (No Response)

5. Is the manuscript presented in an intelligible fashion and written in standard English?

Reviewer #1: Yes

Reviewer #2: Yes

Reviewer #3: (No Response)

6. Review Comments to the Author

Reviewer #1: The authors have made substantial changes on the manuscript, which reviews recent studies on cluster randomized trials (CRTs) with unequal clusters. The revised manuscript now has a clearer structure with sections of trial type, outcome type and design/analysis aspect. The supporting information also provide detailed literature tables that align with the manuscript sections. These changes help readers find relevant material for their uses in CRTs. The authors have addressed my comments on elucidating several methods in the manuscript and also presenting more comprehensive statistics of parameters of each paper in the supporting information tables.

However, the revised manuscript still does not address my previous concern that there is a lack of connections when reviewing the recent papers. Rather than grouping and comparing papers, the authors seem to list and describe the papers that fall into the combination of specific trial type, outcome type and design/analysis aspect. I agree with Reviewer 2 that the review structure follows a one-by-one listing and summarizing process, instead of connecting or comparing the papers.

Although the authors added some summarizing sentences that gives general information on the quantity of papers falling into a category, I expect summaries on the papers described and the connections among them. I agree with the authors that there might not be a general conclusion of papers with very different settings, but the summaries and the connections/transitions can also be an introduction, a view or a comment. For example, in the analysis aspect of continuous outcomes of parallel-arm CRTs, the authors can describe the method trend (GEE, mixed models or others) in this part before going to each paper with its detail. In addition, they can also make more comments on the transitions of papers when they share similar methods, objectives, or conclusions. After all, why are the papers grouped in the same paragraph besides the fact that they fall into the same trial-outcome-design/analysis combination? Can their connections be presented and reviewed more smoothly?

Reviewer #2: (No Response)

Reviewer #3: (No Response)

7. PLOS authors have the option to publish the peer review history of their article (what does this mean?). If published, this will include your full peer review and any attached files.

Reviewer #1: No

Reviewer #2: No

Reviewer #3: No

---

## [Author Response · Author response to Decision Letter 1]

17 Jun 2021

Please find attached for details.

---

## [Decision Letter · Decision Letter 2]

16 Jul 2021

Methods for dealing with unequal cluster sizes in cluster randomized trials: A scoping review

PONE-D-20-23827R2

Dear Dr. Zhan,

We’re pleased to inform you that your manuscript has been judged scientifically suitable for publication and will be formally accepted for publication once it meets all outstanding technical requirements.

Kind regards,

Rafael Sarkis-Onofre

Academic Editor

PLOS ONE

Additional Editor Comments (optional):

Reviewers' comments:

Reviewer's Responses to Questions

**Comments to the Author**

1. If the authors have adequately addressed your comments raised in a previous round of review and you feel that this manuscript is now acceptable for publication, you may indicate that here to bypass the “Comments to the Author” section, enter your conflict of interest statement in the “Confidential to Editor” section, and submit your "Accept" recommendation.

Reviewer #1: All comments have been addressed

2. Is the manuscript technically sound, and do the data support the conclusions?

Reviewer #1: Yes

3. Has the statistical analysis been performed appropriately and rigorously? 

Reviewer #1: Yes

4. Have the authors made all data underlying the findings in their manuscript fully available?

Reviewer #1: Yes

5. Is the manuscript presented in an intelligible fashion and written in standard English?

Reviewer #1: Yes

6. Review Comments to the Author

Reviewer #1: (No Response)

7. PLOS authors have the option to publish the peer review history of their article (what does this mean?). If published, this will include your full peer review and any attached files.

Reviewer #1: No

---

## [Editor Report · Acceptance letter]

21 Jul 2021

PONE-D-20-23827R2 

Methods for dealing with unequal cluster sizes in cluster randomized trials: A scoping review 

Dear Dr. Zhan:

I'm pleased to inform you that your manuscript has been deemed suitable for publication in PLOS ONE. Congratulations! Your manuscript is now with our production department. 

Kind regards, 

on behalf of

Dr. Rafael Sarkis-Onofre 

Academic Editor

PLOS ONE